# Electron probe microanalysis of the elemental composition of phytoliths from woody bamboo species

**Shuhui Tan**[1], **Rencheng Li**[1,2]*, **Richard S. Vachula**[3], **Xinyue Tao**[1], **Mengdan Wen**[1], **Yizhi Liu**[1], **Haiyan Dong**[1], **Lintong Zhou**[1]

**1** College of Earth Science, Guilin University of Technology, Guilin, China, **2** Guangxi Key Laboratory of Hidden Metallic Ore Deposits Exploration, Guilin, China, **3** Department of Geology, College of William and Mary, Williamsburg, VA, United States of America

* lirencheng98@gmail.com

**Data Availability Statement:** All relevant data are within the paper and its Supporting Information files.

## Abstract

Electron probe microanalysis (EPMA) is promising for accurately determining elemental components in micro-areas of individual phytolith particles, interpreting compositional features and formation mechanisms of phytoliths in plants, identifying archeological and sedimental phytolith. However, the EPMA method of analyzing mounted slide phytoliths has not well been defined. In this study, we attempted different EPMA methods to determine the elemental compositions of phytoliths in mounted slides. Direct analysis of carbon (DAC) with other elements in phytolith could obtain abnormally high total values and carbon values. The method of carbon excluded in measuring elements (non-carbon analysis (NCA)) was feasible to obtain elemental compositions in phytolith. The NCA method was conducive to obtain the factual elemental compositions of an individual phytolith (morphotype) when the carbon content of phytolith was relatively low. The EPMA results of phytoliths from 20 bamboo species (three genera) showed that phytolith was dominantly composed of $SiO_2$ but also included low contents of diverse other elements. The EPMA of phytoliths can provide the elemental composition of micro-areas of an individual phytolith particle. The elemental compositions of phytolith varied with their morphotypes, the genera and ecotype of bamboos. The EPMA of elemental compositions in phytoliths is a potential tool to study the formation mechanism of phytoliths, plant taxonomical identification, archaeological and paleoenvironmental reconstruction.

## Introduction

Phytoliths are naturally occurring silica bodies formed within living plants by a biomineralization process. Monosilicic acid, $Si(OH)_4$, taken up by plant roots along with other elements occurring in soil solution, is subsequently deposited in and among cells of the plant husk, leaf, stem and root [1]. Some of these deposits can replicate the morphology of the living cells, producing distinctive and characteristic morphologies. After plants decay, the silica deposited in

**Funding:** This work is supported by grants from the National Natural Science Foundation of China (Grant Nos. 41867058). Recipient: RC Li. The funders had no role in study design, data collection and analysis, decision to publish, or preparation of the manuscript.There was no additional external funding received for this study.

**Competing interests:** The following statement is declared in regard to our manuscript submission of Electron probe microanalysis of the elemental composition of phytoliths from woody bamboo species: All authors have read and approved this version of the article, and due care has been taken to ensure the integrity of the work. No part of this paper has been published or submitted elsewhere. No conflict of interest exists in the submission of this manuscript. There was no additional external funding received for this study

plant tissues is released into the soil in the form of plant microfossils called phytoliths [2–4]. Phytolith morphology is normally different with their produced cell types, tissue and plant taxonomy [1, 5–7]. Thus, phytolith morphology, size and assemblage have been useful in plant taxonomy and ecotype differentiation [6, 8–10], deciphering the mechanism of phytolith formation in plants, understanding taphonomy in soils and sediments [11–14], studying cultivated plant origin, utilization and dispersal [15–19], and paleoenvironmental reconstruction [2, 5, 13, 20–25].

Compared with usual morphological analysis, the study of phytolith composition is relatively underdeveloped. Recently, there has been a growing tendency in phytolith research to move beyond morphology and use phytolith composition as a proxy of its formation, taxonomical origin and environmental context [26]. Various techniques have been used in phytolith composition analysis [1, 14]. Gas chromatography (GC), gas chromatography-mass spectrometry (GC-MS), and $^{13}C$ nuclear magnetic resonance spectroscopy (NMR) were used to determine organic matter in phytolith [27–29]. Chemical functional groups and molecules can be measured by Fourier transform infrared spectroscopy (FTIR) [29–31]. Raman spectroscopy was undertaken using microprobe analysis to reveal the molecular skeleton of phytoliths [29, 32, 33]. The elemental composition and distribution within phytoliths have been examined by scanning electron microscopy-energy dispersive X-ray analysis (SEM-EDX) [18, 30, 34–36], neutron activation analysis (NAA) [37], inductively coupled plasma-atomic emission spectrometry (ICP-AES) [1], and electron probe microanalysis (EPMA) [38, 39].

Based on these analytical efforts, the chemical compositions of phytoliths have begun to be revealed. $SiO_2$ and $H_2O$ are the basic component of phytolith [1, 5]. Recent studies reported that 0.2–5.8% of organic matter can be occluded during the formation of phytoliths [12, 40–42]. Normally cell wall phytoliths can contain larger amounts of carbon than lumen phytoliths. According to Hodson's [43] calculation, the cell wall phytoliths investigated by Perry et al. [44] are around 25% carbon. Parr and Sullivan [45] used an indirect method and estimated percent content in cell wall phytoliths was 10.12% for sugarcane and 3.37% for sorghum. Other datasets, however, showed PhytOC (phytolith-occluded carbon) of 0.1–0.5% of the dry weight [46–49]. Some researches indicate that PhytOC is composed of lipid, protein, carbohydrates and lignin compounds [23, 27, 28, 33]. DNA is absent or not routinely recoverable in a random assemblage of siliceous phytoliths. There is still a possibility that small amounts of DNA are present in specific phytolith types [50]. Phytolith contains various inorganic mineral elements including major elements, such as Si, Al, Ca, N, P, K, and trace elements such as Pb, Fe, Mn, Zn, Cu, As, Ti and so on [1, 4, 26, 37, 51–54]. In addition, the stable and/or isotopic composition of Si, O and C in phytolith was explored to relate with the development of phytoliths in plant, plant growth environmental conditions and geological dating [4, 26].

In summary, these analyses normally involve measurement of phytolith aggregation, and yield average chemical compositions of phytolith aggregates. The use of the electron probe microanalyser and/or energy dispersive X-ray analyzer (EDS) has added a method to study the elemental compositions of micro-areas in individual phytolith particles [35, 38, 55–57]. Especially the electron probe microanalyzer equipped with WDS is promising for more accurately determining elemental components in phytoliths. The electron probe microanalysis of silica in plant tissues and/or certain phytolith morphologies is conducive to interpret physiological function and formation mechanisms of phytolith [4, 38]. However, previous microscopic detections of silica were normally taken in fresh plant tissues [38, 58], and seldom in phytolith particles extracted from plants, soils and sediments [39]. X-ray counts in the epidermal peels is the only qualitative assessments of silicon because the variable density of heterogeneous biological material can cause the scanning beam penetration to vary and the uneven surface of the epidermal peels is subject to topographical variations in X-ray take-off angle [38]. Despite the microscopic

detection of silica in the sectional material of plant tissues, the X-ray intensities are not directly proportional to elemental concentration because all these factors have not been completely eliminated. Although EPMA was conducted for phytolith particles in mounted slides [39], abnormal C values were obtained in some samples, probably due to carbon films on the phytolith sample. The EPMA results are probably influenced by one unanalyzed element because it was not input to ZAF matrix corrections [59]. Thus, it is necessary and crucial to study the EPMA method and obtain authentic elemental compositions of phytolith particles.

In this study, the phytoliths from 20 bamboo species were extracted using a microwave digestion method and the elemental compositions of these phytoliths in mounted slides were examined by an electron probe microanalyzer using different methods. The following research questions are addressed: (i) What method is suitable for EPMA of phytoliths? (ii) How can EPMA be conducted to obtain stable and reliable elemental compositions in phytoliths? (iii) What are the EPMA results of a phytolith particle?

## Materials and methods

### Sampling site

Guilin Botanical Garden is located in the Yanshan District of Guilin City, in the subtropical monsoon climate region of southwest China (Fig 1), and is affected by both the SW maritime monsoon from the Indian Ocean and the SE maritime monsoon from the western Pacific Ocean. It has an annual mean temperature of 18.8˚C, precipitation of 1874 mm, and sunshine of 1670 h. The highest and lowest monthly mean temperatures are 27.7˚C in July and 8.4˚C in January, respectively [60]. The vegetation zone is subtropical evergreen broad-leaved forest. Gramineae grass is widely distributed, with Bambusoideae, Panicoideae and Chloridoideae species dominating, and relatively few Pooideae species distributed [36, 61]. The soil in Guilin Botanical Garden is characterized by laterite developed on carbonate bedrock.

### Sampling and analysis

**Leaf sampling.** About 50g of senescent leaves of 20 Bambusoideae species spanning three genera and two ecotypes (*Phyllostachys* (monopodial scattering), *Dendrocalamus* (sympodial caespitose) and *Bambusa* (sympodial caespitose)), were collected from Guilin Botanical Garden in December 2019 (Table 1). The leaves of *Dendrocalamus ronganensis* grown on top of a karst hill (outside of Guilin Botanical Garden) in Guilin City were also collected in August 2019. The sampling method followed Li et al. [60]. To minimize the differences in growth conditions that might affect leaf phytolith morphology and assemblages, whole, fully developed/senescent bamboo leaf samples were collected from individual plants with similar growth direction and height. The leaves of *D. ronganensis* were obtained earlier to try EPMA using three methods. The 20 bamboo species used to study taxonomical classification were collected from within the same 500 m radius plot in Guilin Botanical Garden.

**Phytolith extraction.** Leaves were thoroughly washed with ultrapure water after an ultrasonic bath for 15 mins, dried at 75˚C for 48 h, cut into small pieces (size:1mm×1mm), and mixed to ensure sample homogeneity. The phytoliths were extracted using a microwave digestion method [42] and Walkley-Black type digest [62]. A 1g subsample of the mixed leaf pieces was placed into a microwave digestion tank. A total of about 3-5g leaf samples were processed. Each phytolith extraction solution used 0.8 mol/L potassium dichromate to examine the extraneous organic materials outside of the phytolith particles [12]. The organic matter in the matrix containing the phytolith was thoroughly removed by ensuring the color of solution would not change within 5 mins [12]. Each extraction was merged together as a sample for EPMA.

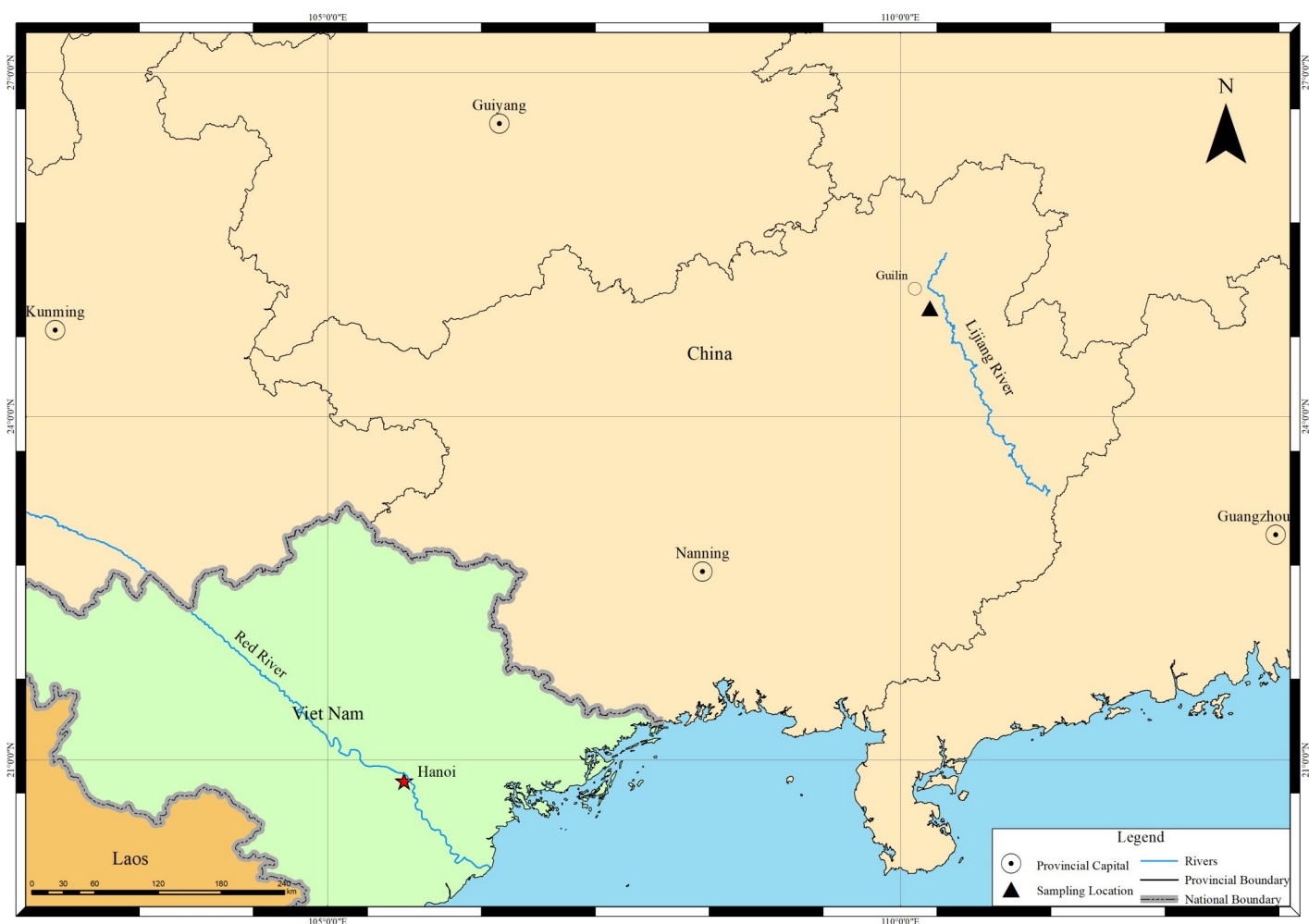

**Fig 1. Map of sampling location.**

## PhytOC measurement

PhytOC content was determined by using alkali dissolution spectrophotometry [63]. The summarized protocol is as follows: (a) approximately 0.01 g of phytoliths was weighed in 10-mL plastic tubes; (b) 0.5 mL NaOH (10mol/L) was added to the tubes and allowed to stand for 12 h, before being transferred into 50-mL centrifuge tubes; (c) 1 mL $K_2Cr_2O_7$ solution (0.8 mol/L) and 4.6 mL concentrated sulfuric acid ($H_2SO_4$) were added to the centrifuge tubes which were then heated in a water bath to 98˚C for 1 h; (d) cooled, distilled water was added to achieve 25 mL solution, and then it was transferred into a 50-mL centrifuge tube, and then centrifuged at 2500 rpm for 10 mins; (e) the absorbance of the supernatant was measured at 590 nm by using a spectrophotometer, and used to calculate the PhytOC content using a calibration curve.

The calibration curve and associated regression equation were established as follows: (a) 0, 0.5, 1.0, 1.5, and 2.0 mL of $C_6H_{12}O_6$ standard solution were respectively added to plastic tubes; (b) 1 mL $K_2Cr_2O_7$ solution (0.8 mol/L) and 4.6 mL concentrated $H_2SO_4$ were placed in the plastic tubes and then heated in a water bath at 98˚C for 1 h; (c) cooled, distilled water was added to achieve 25 mL of the solution, and then it was transferred into a 50-mL centrifuge

**Table 1. Sampling information.**

| Genus | Species | Ecotype | Sampling site |
|---|---|---|---|
| Dendro-calamus | Dendrocalamus ronganensis | sympodial caespitose | karst hill |
| | Dendrocalamus brandisii (Munro) Kurz | | garden |
| | Dendrocalamus farinosus (Keng et Keng. f.) Chia et H. L. Fung | | |
| | Dendrocalamus minor (McClure) Chia et H. L. Fung | | |
| | Dendrocalamus giganteus Munro | | |
| | Dendrocalamus pachystachys Hsueh et D. Z. Li | | |
| | Dendrocalamopsis vario-striata (W. T. Lin) Keng f. | | |
| | Dendrocalamus sapidus Q. H. Dar et D. Y. Huang | | |
| Bambusa | Bambusa eutuldoides McClure | sympodial caespitose | garden |
| | Bambusa chungii McClure | | |
| | Bambusa multiplex (Lour.) Raeusch. ex Schult. 'Fernleaf' R. A. Young | | |
| | Bambusa textilis McClure | | |
| | Bambusa remotiflora Kuntze | | |
| | Bambusa blumeana J. A. et J. H. Schult. f | | |
| | Bambusa contracta Chia et H. L. Fung | | |
| | Bambusa albo-lineata Chia | | |
| Phyllo-stachys | Phyllostachys sulphurea (Carr.) A. et C. Riv | monopodial scattering | garden |
| | Phyllostachys heterocycla (Carr.) Mitford cv. Pubescens | | |
| | Phyllostachys nigra (Lodd. ex Lindl.) Munro | | |
| | Phyllostachys sulphurea (Carr.) A. 'Viridis' | | |
| | Phyllostachys praecox C. D. Chu et C. S. Chao 'Prevernalis' | | |

tube, then centrifuged at 2500 rpm for 10 mins; (d) the absorbance of the supernatant was measured at 590 nm by using a spectrophotometer; (e) the standard calibration curve was plotted and the regression equation was established.

## Electron probe microanalysis

**Mounted slide preparation.** Approximately 0.1 g phytolith particles was mixed homogenously with epoxy resin glue and mounted on microscope slides. First, the phytolith and epoxy resin mixture was coarsely ground using automatic mill. Then, the covering was ground finely to about 0.03 mm thickness using a 320 mesh fine grinding mill (with W14 motor). Subsequently, the phytolith mixture covering was rubbed using polishing solution, and then was polished to smoothness with a polisher. The polished section was wiped with ethyl alcohol and cleaned with water. Finally, the phytolith slice was sprayed with carbon to prepare for EPMA.

**Electron probe microanalysis.** The quantitative elemental compositions in phytolith were determined using a JEOL JXA-8230 electron probe microanalyzer (EPMA) at Guilin University of Technology. The operating beam conditions were as follows: accelerating voltage 15 kV, beam current 20 nA, and focused beam diameter 2 or 5 μm. The BSE images were mainly obtained with an accelerating voltage of 15 kV.

Natural and synthetic standards were used, and matrix corrections were based on ZAF procedures. All data were processed with the ZAF correction procedure. The backscatter electron (BSE) images were obtained with an accelerating voltage of 15 kV. Both natural and synthetic standards were used. Not less than 10 elements (Na, Si, Al, Mg, Ca, Mn, Cr, P, K and Fe) were chosen for EPMA. The typical detection limits for oxides of most elements are better than 0.02 wt.%. All data were processed with the ZAF correction procedure supplied by the JEOL microprobe. Three experimental strategies were applied and attempted to

phytolith samples during EPMA: (i) Direct analysis of carbon (DAC, C was analyzed) method. For this method C with 10 elements (Na, Si, Al, Mg, Ca, Mn, Cr, P, K and Fe) were chosen for EPMA; (ii) Non-carbon analysis (NCA, C was not analyzed) method. For this method, compared to DAC, 10 elements (Na, Si, Al, Mg, Ca, Mn, Cr, P, K and Fe) except C were chosen for EPMA; (iii) Fixed carbon for CAL model (Fix-C, assuming a C value to calculate matrix correction (ZAF) calibration, the assuming C values equal the content of PhytOC). For this method, calibration (CAL) model fixed C values, 10 elements (Na, Si, Al, Mg, Ca, Mn, Cr, P, K and Fe) were chosen for EPMA. Overall, 20–30 phytolith particles included bulliform cell, short cell and long cell phytoliths were chosen for EPMA, respectively. The average and total content of elements for GSSCP, ELOGATE, BULLIFORM FLABELLATE phytoliths, and phytolith aggregations (including all phytolith morphotypes) were calculated. Phytoliths from *D. ronganensis* leaves were taken for EPMA by using three methods. Two points on each selected individual phytolith from the other 20 species were analyzed with the NCA and Fix-C models, respectively.

## Statistical analyses

The elemental compositions of phytolith were conducted hierarchical cluster analysis to investigate the affinities among 20 woody bamboo species. Correlation analyses were performed to check the relationships between EPMA results of phytolith obtained by using the NCA and Fix-C model. All statistical analyses were performed by SPSS 19.0 software.

## Results

### EPMA results from three different methods

Phytolith particle were identified by EDS analysis, which revealed Si, O and C were the major constituent elements of particles before EPMA was undertaken (Fig 2). The electron probe microanalyzer measured the elemental composition of specific micro-areas of an individual phytolith morphotype (Table 2 and Fig 3). K, Ca, Mg, Na, Mg, Cr, Fe, Mn and Al in phytolith particles exceeded the limit of detection. The EPMA results of phytoliths from *D. ronganensis* using the DAC method showed abnormally high total values and carbon values. The total and

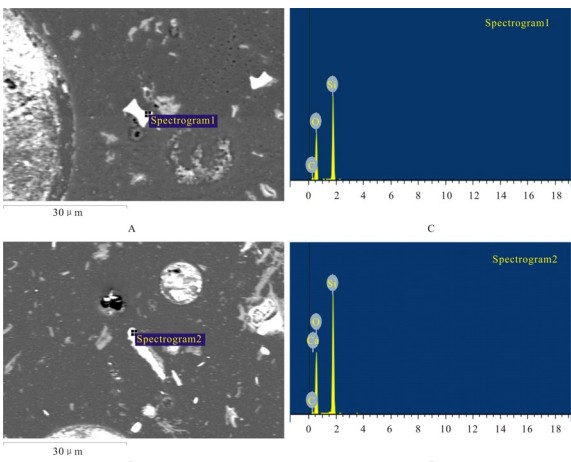

**Fig 2. Energy dispersive X-ray analyzer (EDS) analysis revealing constituent elements of phytoliths.** Each element is identified by characteristic X-ray peaks of specific energy. A, B, Back-scattered electron imaging (BSE). C, D present C, Si, O in SADDLE and ELONGATE phytolith, respectively.

**Table 2. The EPMA results of each numbered micro-area of major phytolith morphotypes (unit: %).**

| Point numbered | EPMA results | | | | | | | | | | | |
|---|---|---|---|---|---|---|---|---|---|---|---|---|
| | SiO₂ | Na₂O | Al₂O₃ | MgO | CaO | MnO | Cr₂O₃ | P₂O₅ | K₂O | FeO | Total | C+H₂O* |
| 1 | 91.02 | 0.05 | 0.03 | 0.02 | 0.04 | 0.00 | 0.01 | 0.00 | 0.09 | 0.00 | 91.26 | 8.74 |
| 2 | 92.06 | 0.06 | 0.02 | 0.03 | 0.02 | 0.00 | 0.05 | 0.01 | 0.09 | 0.00 | 92.34 | 7.66 |
| 3 | 90.70 | 0.06 | 0.05 | 0.09 | 0.37 | 0.03 | 0.03 | 0.00 | 0.41 | 0.00 | 91.74 | 8.26 |
| 4 | 89.68 | 0.07 | 0.03 | 0.10 | 0.37 | 0.02 | 0.03 | 0.02 | 0.40 | 0.01 | 90.72 | 9.28 |
| 5 | 88.82 | 0.02 | 0.03 | 0.05 | 0.09 | 0.01 | 0.06 | 0.06 | 0.03 | 0.00 | 89.15 | 10.85 |
| 6 | 90.00 | 0.02 | 0.04 | 0.05 | 0.08 | 0.06 | 0.00 | 0.00 | 0.05 | 0.00 | 90.31 | 9.69 |
| 7 | 84.89 | 0.08 | 0.02 | 0.03 | 0.02 | 0.03 | 0.09 | 0.03 | 0.24 | 0.01 | 85.43 | 14.57 |
| 8 | 88.19 | 0.02 | 0.06 | 0.02 | 0.04 | 0.02 | 0.06 | 0.00 | 0.16 | 0.00 | 88.56 | 11.44 |
| 9 | 89.50 | 0.04 | 0.12 | 0.01 | 0.04 | 0.08 | 0.03 | 0.03 | 0.28 | 0.01 | 90.15 | 9.85 |
| 10 | 76.08 | 0.07 | 0.05 | 0.13 | 0.10 | 0.16 | 0.02 | 0.00 | 0.22 | 0.03 | 76.85 | 23.15 |
| 11 | 87.81 | 0.04 | 0.01 | 0.00 | 0.06 | 0.00 | 0.00 | 0.00 | 0.13 | 0.00 | 88.05 | 11.95 |
| 12 | 88.36 | 0.05 | 0.00 | 0.00 | 0.04 | 0.00 | 0.07 | 0.00 | 0.15 | 0.00 | 88.66 | 11.34 |
| 13 | 89.82 | 0.04 | 0.02 | 0.01 | 0.08 | 0.02 | 0.10 | 0.00 | 0.08 | 0.00 | 90.15 | 9.85 |

C+H₂O*: $C+H_2O = 100-SiO_2-Na_2O-Al_2O_3-MgO-CaO-MnO-Cr_2O_3-P_2O_5-K_2O-FeO$

carbon values ranged from 86.8 to 138.5% (average 105.68%), and 6.0 to 53.2% (average 19.25%), respectively (Table 1 and Fig 4).

Fixed carbon values were derived from the PhytOC content of 20 bamboo species leaves (Table 4). The total values ranged from 83.27 to 97.04% (average 86.0%), and from 77.3 to 98.3% (average 91.12%) when NCA and Fix-C model were used, respectively (Table 3 and Fig 4). A positive correlation ($R^2 = 0.94$, n = 9, p<0.01) existed between the contents of elements determined (except for content of SiO₂, because its overwhelming high content will cause $R^2 = 1$) in phytolith from *D. ronganensis* leaves obtained by using NCA method and Fix-C model (Fig 5A). The same positive correlations (average $R^2 = 0.88$) existed between 9 elemental compositions of phytolith for all of these species. The EPMA results of phytoliths were comparable for these bamboo species when either the NCA or Fix-C model were used. All of the total values were lower than 100%. Positive correlations ($R^2 = 0.32$, n = 20, p<0.01) existed between the SiO₂ content derived from these two methods (Fig 5B). However, the total values from the NCA method were generally less than the fixed C values than those from the fixed C model. The difference between the Total values from the NCA and fixed C model were positively correlated with fixed C values ($R^2 = 0.57$, n = 20, p<0.01) (Table 4 and Fig 5C).

## Elemental compositions of phytolith

The elemental compositions in phytoliths from 20 bamboo species determined by electron probe microanalyzer using NCA and Fix-C methods showed that (1) the total values ranged from 83.11 to 87.49%, and 85.16 to 92.25%, (2) SiO₂ was the dominant component and varied from 82.62 to 87.06%, and 83.69 to 86.80%, respectively, and (3) the total amount of other elements (K, Ca, Mg, Na, Cr, Fe, Mn and Al) varied between 0.36 to 0.86%, and 0.39 to 0.90% respectively. The Ca, K, and Al had relatively high contents. The difference between 100% and total values was proposed to be the content of H₂O and/or carbon, as C and H were the dominant light elements in phytoliths [1, 5, 33, 64], and not be examined in this study. Thus, the content of H₂O and C ranged from 12.57 to 16.87% for NCA, and from 7.75 to 13.90% for Fix-

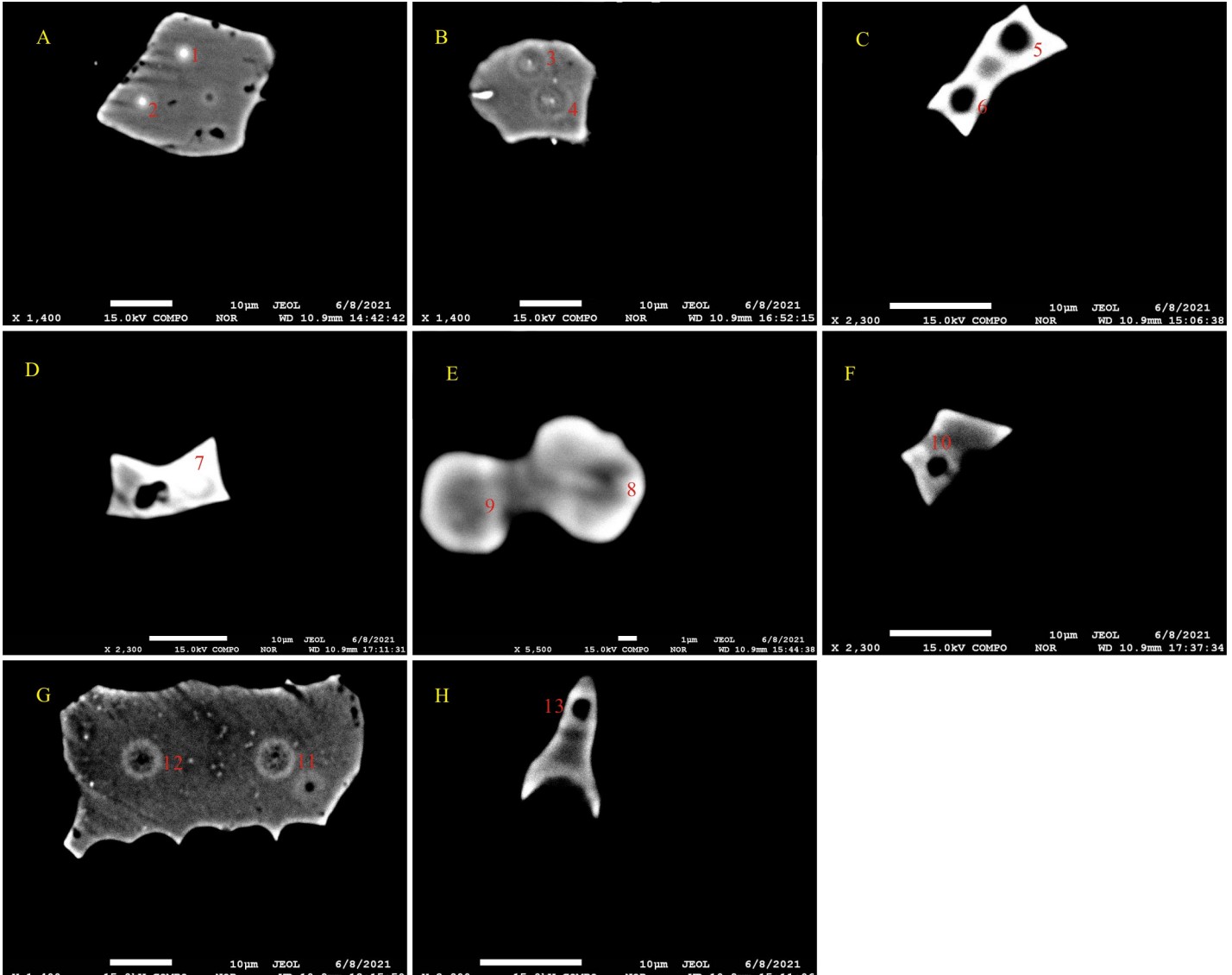

**Fig 3. Back-scattered electron imaging of different phytolith morphotypes.** The point numbered is focused beam hitting micro-areas of an individual phytolith particle. A, B, BULLIFORM FLABELLATE; C, D, SADDLE; E, BILOBATE; F, RONDEL; G, ELONGATE; H, ACUTE BULBOSUS.

C method (Table 4). The values of elemental compositions in phytolith varied with morphotypes and the plant species which produced the phytoliths (Tables 2 and S1A).

## Taxonomical and ecotypical significance of phytolith elemental compositions

The average $SiO_2$ content and total values of phytoliths varied between genera or ecotype, increasing from *Bambusa* (sympodial caespitose), *Dendrocalamus* (sympodial caespitose), to *Phyllostachys* (monopodial scattering), and also varied between bamboo genera or ecotype (Table 4 and Fig 6). The relative abundances of elemental compositions in phytoliths do not show taxonomical significance at genus level.

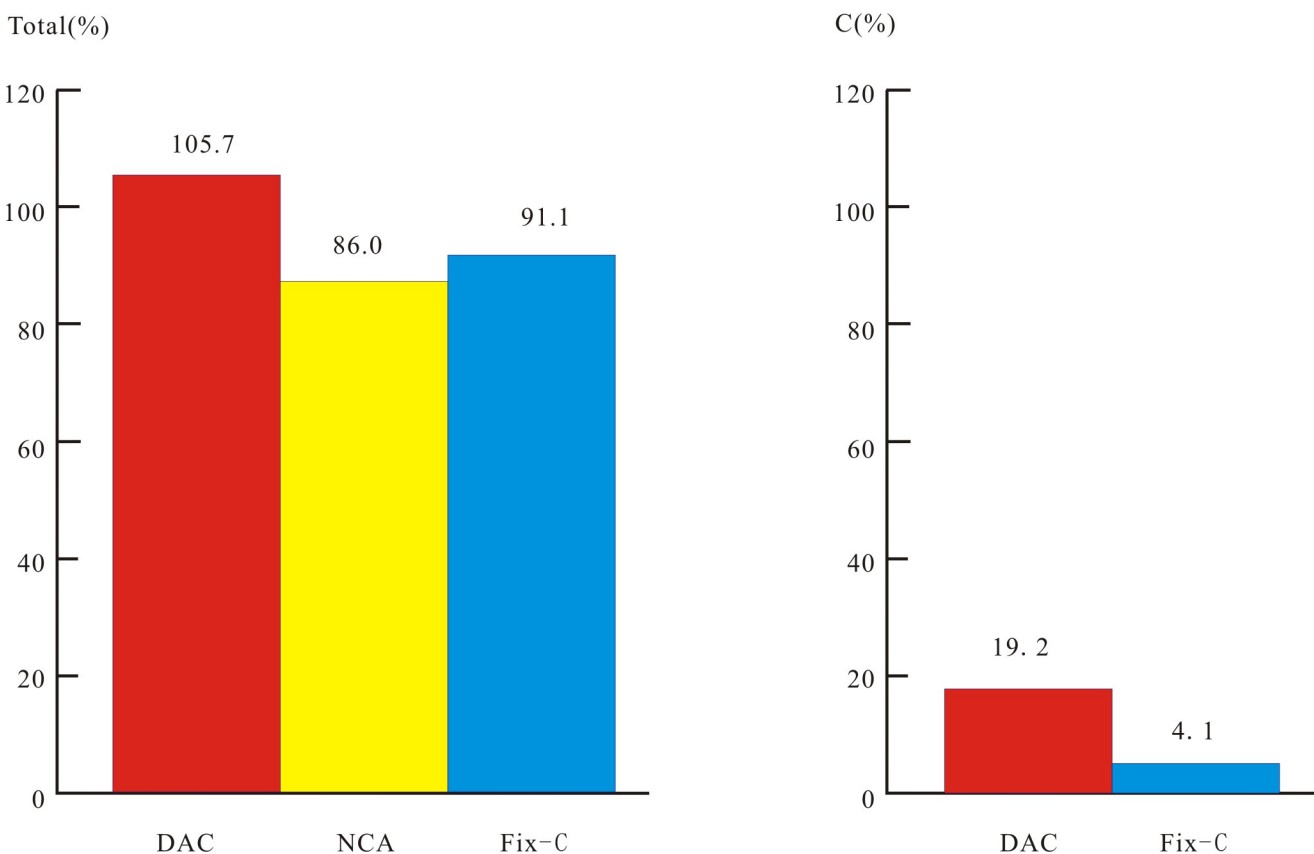

**Fig 4. The difference between the average total values of phytolith EPMA by using three model.** Note: Phytoliths were extracted from *D. ronganensis* leaves. The total number of phytolith particles used for EPMA were 24. Fix-C carbon value was equivalent to the PhytOC value.

## Discussion

### A feasible method of EPMA for mounted slide phytolith samples

The electron probe microanalyzer is an ideal instrument to determine the chemical composition in a micro-area of a specimen. The characteristic X-ray corresponding to the chemical composition of the specimen is obtained. The wavelength and intensity of the characteristic X-ray are subsequently analyzed by wavelength-dispersive spectroscopy (WDS) to qualify and quantify the chemical elemental composition of the surface [65]. Accurate analysis for carbon by EPMA of C-coated insulating samples, often in the presence of C-based vacuum-pump oils, polishing abrasives, and lubricants, is problematic [59]. Meanwhile, a drawback of EPMA, until recently, was the inability to routinely and precisely analyze light elements because of their low X-ray yield due to their absorption in the specimen, the analyzer crystal and the detector window [66]. The abnormal EPMA total values and carbon content obtained using DAC method probably resulted from carbon as a light element and/or the effect of the sample's carbon coat. So, the DAC method was unable to obtain accurate and precise elemental compositions in phytoliths. The technique of direct analysis of carbon in phytolith samples therefore needs further study.

For all phytolith samples, the elemental compositions obtained by using NCA and Fix-C model were similar and positively correlated, and were very comparable with published data. The total values of EMPA being less than 100% when using the non-carbon analysis and fixed

**Table 3. The elemental content and total values of phytolith determined by EPMA with three methods (unit: %).**

| Species | Method | SiO₂ | Na₂O | Al₂O₃ | MgO | CaO | MnO | Cr₂O₃ | P₂O₅ | K₂O | FeO | C | Total |
|---|---|---|---|---|---|---|---|---|---|---|---|---|---|
| *D. ronganensis* | DAC | 86.04 | 0.02 | 0.06 | 0.05 | 0.12 | 0.02 | 0.02 | 0.01 | 0.09 | 0.01 | 19.25 | 105.68 |
| | NCA | 85.57 | 0.03 | 0.05 | 0.05 | 0.11 | 0.03 | 0.05 | 0.01 | 0.07 | 0.01 | 0.00 | 85.96 |
| | Fix-C | 86.62 | 0.02 | 0.05 | 0.05 | 0.12 | 0.03 | 0.03 | 0.01 | 0.07 | 0.02 | 4.10 | 91.12 |

Each value was averaged from EPMA data of 24 phytolith particles including BULLIFORM FLABELLATE, Grass silica short cell and ELONGATE phytoliths).

DAC = direct analysis of C, NCA = non-carbon analysis, Fix-C = Fixed C model.

C models can be attributed to light elements not being detected [66]. EPMA data show phytolith is mainly composed of $SiO_2$, and but also contain K, Ca, Mg, Na, Cr, Fe, Mn and Al, as

**Table 4. The EPMA (using NCA and Fix-C method) results (elemental compositions and total values (wt.%)) of phytoliths in leaves of bamboo species from three genera.**

| Genus | Species | NCA | | | | Fix-C | | | | [a]R² | [b]C | [c]△Total |
|---|---|---|---|---|---|---|---|---|---|---|---|---|
| | | SiO₂ | Total_N-SiO₂ | Total_N | 100-Total_N | SiO₂ | Total_F-SiO₂ | Total_F | 100-Total_F | | | |
| Dendro-calamus | *Dendrocalamus brandisii* (Munro) Kurz | 85.42 | 0.45 | 85.87 | 14.13 | 85.42 | 0.47 | 88.61 | 11.39 | 0.99 | 2.72 | 2.75 |
| | *Dendrocalamus farinosus* (Keng et Keng. f.) Chia et H. L. Fung | 85.22 | 0.75 | 85.97 | 14.03 | 84.25 | 0.82 | 86.46 | 13.54 | 0.96 | 1.40 | 0.49 |
| | *Dendrocalamus minor* (McClure) Chia et H. L. Fung | 83.27 | 0.55 | 83.82 | 16.18 | 84.67 | 0.65 | 86.72 | 13.28 | 0.94 | 1.40 | 2.90 |
| | *Dendrocalamus giganteus* Munro | 86.18 | 0.60 | 86.79 | 13.21 | 86.75 | 0.50 | 89.69 | 10.31 | 0.98 | 2.44 | 2.90 |
| | *Dendrocalamus pachystachys* Hsueh et D. Z. Li | 83.21 | 0.53 | 83.73 | 16.27 | 82.96 | 0.54 | 85.16 | 14.84 | 0.98 | 1.66 | 1.42 |
| | *Dendrocalamopsis vario-striata* (W. T. Lin) Keng f. | 85.42 | 0.50 | 85.92 | 14.08 | 84.36 | 0.51 | 86.84 | 13.16 | 0.99 | 1.97 | 0.92 |
| | *Dendrocalamus sapidus* Q. H. Dar et D. Y. Huang | 84.50 | 0.45 | 84.94 | 15.06 | 83.82 | 0.40 | 87.57 | 12.43 | 0.98 | 3.34 | 2.63 |
| | **All species (Average)** | **84.75** | **0.55** | **85.29** | **14.71** | **84.61** | **0.56** | **87.29** | **12.71** | **0.97** | **2.13** | **2.00** |
| Bambusa | *Bambusa eutuldoides* McClure | 85.16 | 0.86 | 86.02 | 13.98 | 84.84 | 0.90 | 89.52 | 10.48 | 0.96 | 3.78 | 3.50 |
| | *Bambusa chungii* McClure | 84.78 | 0.58 | 85.35 | 14.65 | 86.01 | 0.43 | 88.78 | 11.22 | 0.92 | 2.34 | 3.43 |
| | *Bambusa multiplex* (Lour.) Raeusch. ex Schult. 'Fernleaf' R. A. Young | 84.80 | 0.48 | 85.28 | 14.72 | 85.92 | 0.45 | 89.34 | 10.66 | 0.90 | 2.97 | 4.06 |
| | *Bambusa textilis* McClure | 86.36 | 0.53 | 86.89 | 13.11 | 86.05 | 0.65 | 89.24 | 10.76 | 0.67 | 2.53 | 2.34 |
| | *Bambusa remotiflora* Kuntze | 83.74 | 0.84 | 84.57 | 15.43 | 85.89 | 0.66 | 88.33 | 11.67 | 0.97 | 1.78 | 3.75 |
| | *Bambusa blumeana* J. A. et J. H. Schult. f | 85.41 | 0.54 | 85.95 | 14.05 | 85.53 | 0.48 | 89.04 | 10.96 | 0.96 | 3.03 | 3.09 |
| | *Bambusa contracta* Chia et H. L. Fung | 82.62 | 0.49 | 83.11 | 16.89 | 83.82 | 0.44 | 86.10 | 13.90 | 0.98 | 1.84 | 2.99 |
| | *Bambusa albo-lineata* Chia | 84.60 | 0.60 | 85.20 | 14.80 | 83.69 | 0.81 | 87.03 | 12.97 | 0.62 | 2.53 | 1.83 |
| | **All species (Average)** | **84.68** | **0.61** | **85.30** | **14.70** | **85.22** | **0.60** | **88.42** | **11.58** | **0.87** | **2.60** | **3.13** |
| Phyllo-stachys | *Phyllostachys sulphurea* (Carr.) A. et C. Riv | 83.51 | 0.81 | 84.32 | 15.68 | 84.15 | 0.67 | 90.02 | 9.98 | 0.83 | 5.19 | 5.70 |
| | *Phyllostachys heterocycla* (Carr.) Mitford cv. Pubescens | 84.17 | 0.36 | 84.52 | 15.48 | 86.59 | 0.52 | 91.89 | 8.11 | 0.24 | 4.78 | 7.37 |
| | *Phyllostachys nigra* (Lodd. ex Lindl.) Munro | 83.17 | 0.61 | 83.77 | 16.23 | 84.22 | 0.53 | 86.96 | 13.04 | 0.90 | 2.22 | 3.19 |
| | *Phyllostachys sulphurea* (Carr.) A. 'Viridis' | 86.41 | 0.49 | 86.91 | 13.09 | 84.84 | 0.58 | 87.88 | 12.12 | 0.92 | 2.47 | 0.97 |
| | *Phyllostachys praecox* C. D. Chu et C. S. Chao 'Prevernalis' | 87.06 | 0.43 | 87.49 | 12.51 | 86.80 | 0.39 | 92.25 | 7.75 | 0.93 | 5.06 | 4.75 |
| | **All species (Average)** | **84.86** | **0.54** | **85.40** | **14.60** | **85.32** | **0.54** | **89.80** | **10.20** | **0.76** | **3.94** | **4.40** |

Total_N-SiO₂, Total_F-SiO₂ is equal to the weight of other elements. 100-Total_N means the content of H₂O and C, 100-Total_F means the content of H₂O because H and/or C are major elements which could not be determined by EPM analyzer. R² is the square of the Pearson coefficient between the elemental compositions in phytolith from NCA and Fix-C model. [b]C is carbon content of phytolith. [c]△Total is the difference between Total_F and Total_N.

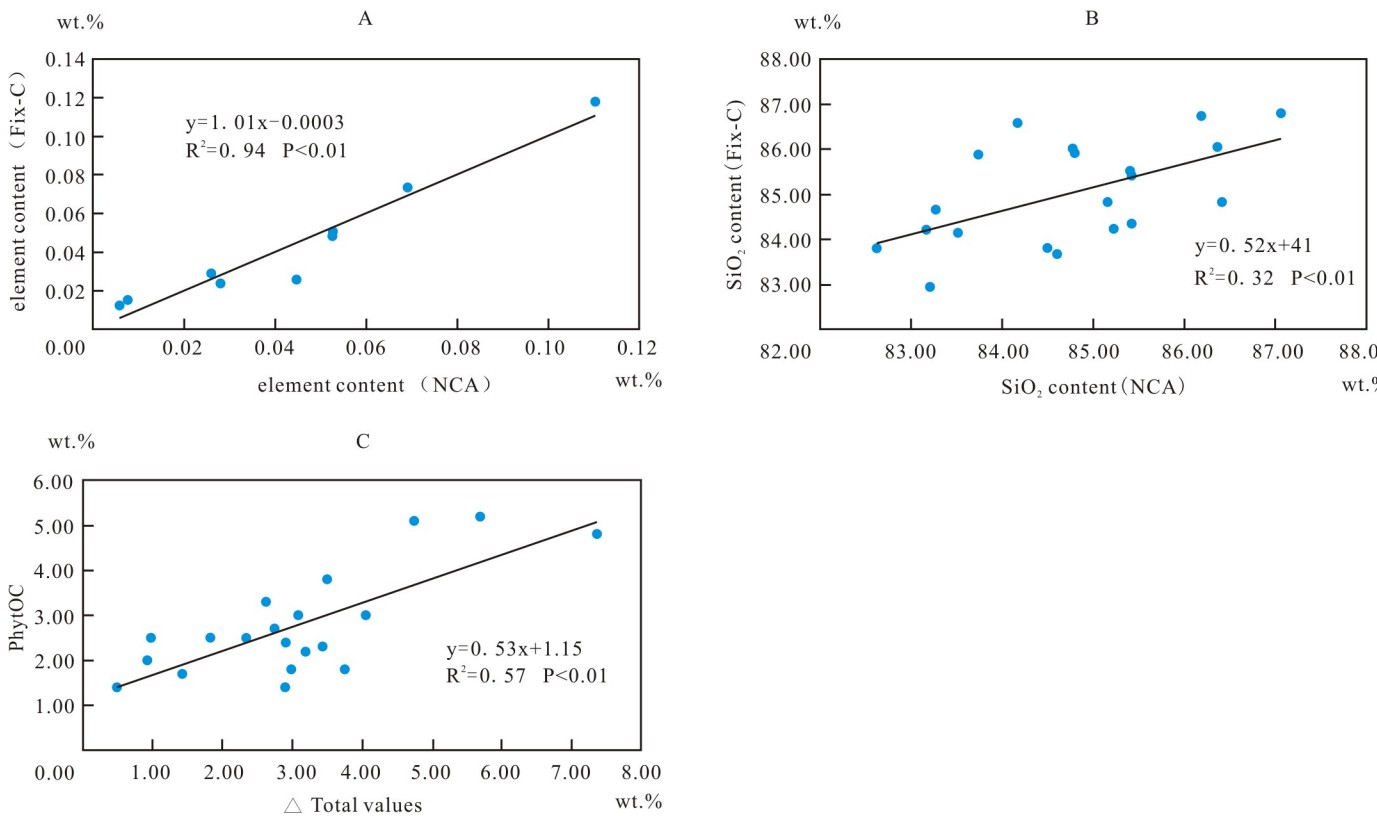

**Fig 5. EPMA results of phytolith are comparative when using non-carbon analysis and the Fix-C model.** A, a positive correlation exists between the contents of 9 elements determined in phytolith particles from *D. ronganensis* leaves; B, the SiO₂ content of phytoliths determined using NCA and Fix-C model correlated positively; C, △Total values (the difference between the total values from NCA and Fix-C model) correlated with fixed C values (PhytOC).

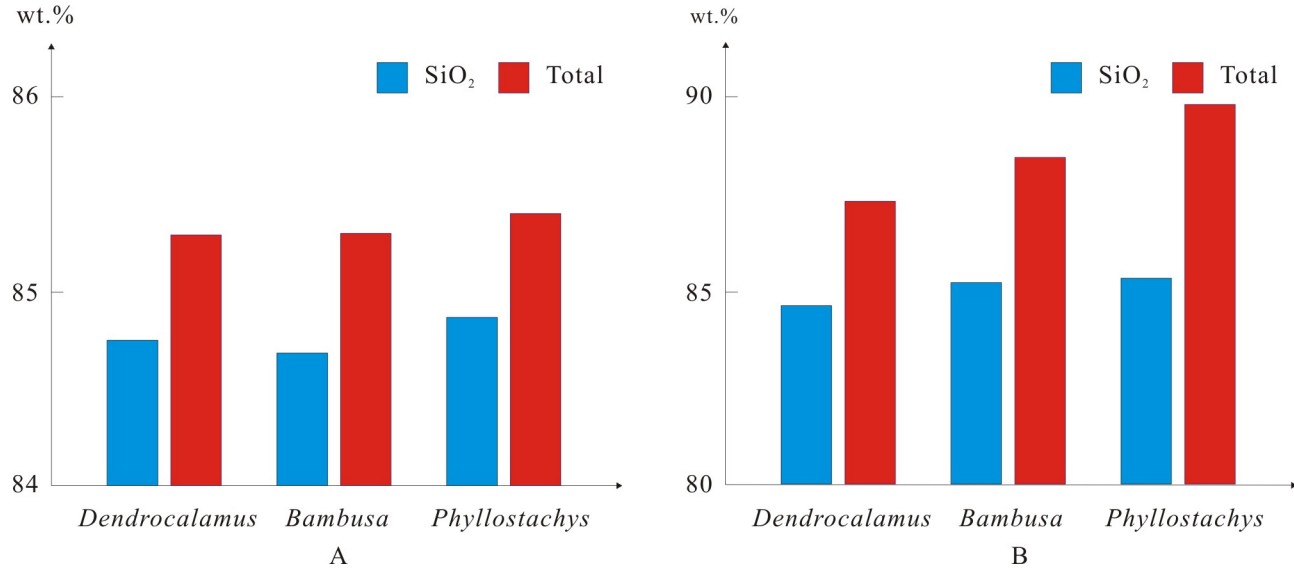

**Fig 6. SiO₂ content and total values of phytolith EPMA varied between bamboo genera and ecotype.** A, NCA (non-carbon analysis) method, B, Fix-C (Fixed C) method.

well as high proportions of $H_2O$ and C. These results are in agreement with measurements made using other techniques. The content of $SiO_2$ and $H_2O$ range from 66–91%, and 0–11% respectively [5, 40]. The concentration of Si analyzed by ICP-AES showed the content of phytolith from different leaves of several rice cultivars ranged from 83.22 to 93.54% (mean 88.47%), and the calculated $H_2O$ content in these phytoliths ranged from 3.86 to 14.46% (mean 9.15%) [1]. The thermogravimetric analysis (TGA) of microwave-assisted digestion phytolith sample showed that the weight loss was 12% and was assumed to be composed mainly of bound water (up to 150˚C) and organic matter [29]. These findings positively correspond to our results, regardless of the differences of vegetation. During the development of a phytolith, elements other than silicon and oxygen might be incorporated into their structures as silica is deposited in cell walls or during the breakdown of the cellular contents. Elements such as C, O, Na, Mg, Si, K, Ca, P, K, Al and Fe were identified in phytoliths in mulberry (*Morus alba*) leaves, rice straw and leaves, and *Than* tree leaves by Scanning electron microscopy (SEM) coupled with an energy dispersive X-ray analyzer (EDS) [35, 36, 56, 57], effectively identifying the presence and formation of phytoliths in plant tissues [36]. The advantage of Energy-dispersive Miniprobe Multielement Analyzer (EMMA-XRF) compared to a SEM coupled with EDX-analysis, is that EMMA-XRF is a quantitative method rather than being semi-quantitative. Light elements (Mg and Al) and heavy elements (Ca, Fe, Cu, Cr, Mn, Zn, Pb, and As) can be detected in the phytoliths of leaves of Ericaceae by EMMA-XRF [51]. Hart [67] analyzed phytoliths from *Actinotus helanthi* (Apiaceae) and *Triodia mitchelli* (Poaceae) for 24 elements (boron, sodium, magnesium, aluminium, phosphorus, calcium, scandium, titanium, vanadium, chromium, manganese, iron, nickel, copper, zinc, gallium, germanium, selenium, ytterbium, zirconium, barium, neodymium, lead and thorium) using inductively coupled plasma source spectroscopy (ICP/MS). Elements like Al, Fe, Mg, Na, Ca, Mn, K and Ti were present in phytoliths within a range of 0.1 to 5.6% of total phytolith mass [1, 68]. Meanwhile, the discrepancies of elemental composition and content between these reported methods are probably derived from the element variations examined, testing techniques, and the plant species.

The EPMA results of phytolith in mounted slide are comparable with the published data, indicating NCA and Fix-C methods are feasible and practical. Considering the results of EPMA by using the non-carbon analysis and fixed carbon model, the positive correlation between these two methods demonstrates that they are suitable for measuring elemental compositions in phytoliths. However, ZAF is a matrix correction factor, required if the sample and standard compositions are significantly different [59]. The Fix-C model assumes that different phytolith morphologies have the same C contents and use the C element content in the ZAF correction. Thus, it is helpful for obtaining accurate measurements of other elements and high total values alike. The NCA method does not include carbon in the ZAF calculation and could influence the accuracy of elemental measurements. Whereas, the NCA method could avoid fixed C with different phytolith morphotypes biases the EPMA results of other elements because of variable occluded carbon between phytolith morphotypes. In this study, the positive correlation between EPMA results derived from these two methods was probably due to low carbon content in phytoliths measured. Compared to the Fix-C method, the NAC method is useful for distinguishing the elements of phytoliths with different morphologies when the carbon content of phytolith is relatively low. Future work needs to study the links between the carbon content and EPMA of elemental compositions in phytoliths.

## The improved techniques on EPMA of phytolith samples

The reliable EPMA results of elemental composition from phytoliths in this study probably depended on several improved operations and techniques. Firstly, increasing the amount of

phytolith powder (more than 100 mg) adhered with epoxy resin on the glass slide guaranteed enough intact individual phytoliths in the polished section to be examined during the EPMA experiment. The adhesive agent epoxy resin might fuse to cause a polished section (even phytolith) to be deformed because the mounted phytolith slide is altered by the beam for some time during the EPMA. An unfixed phytolith particle in the slide (via the deformation of the phytolith slide) would cause the beam missing on phytolith particle to affect the accuracy of the experimental result. Increasing the amount of sample is therefore conducive to shortening the time needed to find phytolith particles for analysis. Secondly, the analysis should be done within a few months of phytolith extraction because extracted phytoliths can easily be eroded and weathered. Phytolith particles could be penetrated and/or broken into pieces by the EPMA beam because of this erosion. Thirdly, the surface of the polished section should be smooth. An uneven surface on the polished section is subject to topographical variations in X-ray take-off angle, and often results in anomalous measurement of X-ray intensities [38]. In addition, selecting the suitable beam diameter based on phytolith size, and clearly aiming the crosshair at a micro-area on the phytolith is necessary to obtain accurate detection and results.

## Elemental compositions affected by phytolith morphotype

The elemental compositions and total values of EPMA varying between the micro-areas and morphotypes of phytoliths analyzed suggest that the compositions in phytolith are inhomogeneous and heterogeneous and may be related to its morphotype (Tables 2 and S1A). These results can be explained by the development and formation of phytoliths during plant growth. A hypothetical model was proposed that plant cell organelles, the nucleus, the chloroplasts and other organic components can be occluded during phytolith formation [69]. Some mineral elements occluded in phytolith have also been reported [1, 35, 37, 39, 51, 52, 54]. Carbon and some mineral elements distributed in homogeneously and unevenly in a phytolith particle has been demonstrated in some plant species [35, 52, 56]. Our EMPA results of phytolith suggest the proportion of carbon and mineral elements might vary with phytolith morphotype. The cell conditions in which silica deposited may result in a variation of the refractive index of plant silica, indicating variation of hydration levels and mineral compositions [51]. The silica matrix is not the same when a phytolith forms in the cell lumen or cell wall. Phytolith deposited in lumen do not appear to be laid down onto a carbohydrate matrix in the same way as the wall deposits [26]. The cuticle might be incorporated into the mineral of epidermal long cells [29]. The thick and lignified walls are probably involved in silica mineralization of long cells. The mineral in long cells (ELONGATE) contains a greater fraction of organic residues [29], The cell occlusions and substances varying with cell types and developing age are assumed to cause the chemical compositions of phytolith to vary with morphotype [4].

## Elemental compositions of phytolith distinguish taxonomical relationships between bamboo genera

The average and total $SiO_2$ content of phytoliths in leaves varied between bamboo genera, suggesting that bamboo species and ecotypes might influence the elemental compositions of phytoliths. Phytoliths in *Phyllostachys* species exhibit higher $SiO_2$ content than those of *Dendrocalamus* and *Bambusa*, which could be interpreted to reflect monopodial scattering species occupying larger land areas and being more conducive to absorb soil Si and increase transpiration compared to sympodial caespitose species. We hypothesize that the higher $SiO_2$ content in phytoliths results from plant leaves obtaining more dissolved Si and suffering stronger transpiration, and this could therefore reflect taxonomical characteristics and relationships. These inferences are supported by the observation that the average length and width of

concave saddle phytolith in bamboo leaves decreased from *Phyllostachys* to *Dendrocalamus* and *Bambusa*, which is likely related to the genetic and ecotypic differentiation [5, 10].

The potential of the chemical composition of phytoliths having taxonomic significance has been reported by some authors. For instance, Hart [67] observed that phytoliths from *Actinotus helanthi* (Apiaceae) and *Triodia mitchelli* (Poaceae) showed very different elemental compositions despite the plants growing in the same soil and environment. Carnelli et al. [70] investigated the chemical composition of phytoliths from 20 species occurring in subalpine and alpine grasslands, heaths, and woodlands on siliceous bedrock, and found only woody species produced a high proportion of phytoliths containing aluminum. They considered this difference to reflect that the chemical compositions of wood and that of herbaceous phytoliths has important implications for the sourcing of phytoliths.

In this study, the elemental compositions of phytoliths do not show the taxonomic significance at the genus level as well as the phytolith assemblage and sizes [9, 10], indicating that the elemental composition of phytoliths is likely more influenced by environmental conditions than the morphotype of phytoliths. It is well known that the phytolith morphology reflects cell morphology within plant tissues [1], which is mainly controlled by genetics [5, 71]. The elemental compositions in phytoliths could be influenced by the cellular environment of silica deposited, the species of plant and the environmental conditions [1, 14, 43]. The difference of elemental compositions of phytoliths among these three genera should be related with bamboo species and their growing environmental condition because the phytolith morphotypes chosen for EMPA were the same. Silica accumulation results from water consumption by the plant, and/or is genetically controlled [7, 71–73]. Our results show the elemental compositions of phytoliths from these three bamboo genera is influenced by both plant growth environmental conditions and plant species and ecotype, and that phytolith elemental composition compared to morphology is more sensitively influenced by plant growing environmental factors.

## Potential implications for EPMA of phytolith

EPMA can provide the elemental compositions of micro-areas of an individual phytolith particle, which can be insightful for interpreting the formation mechanism of phytoliths in plants. It is possible to explore the relationships between phytolith formation, genetic effects and/or growth conditions of plant. Plant cells change physiological functions and elemental compositions with their types. It is reasonable to infer the elemental compositions of phytoliths to distinguish their morphotype, as silica deposited with cytoplasm and organelles in a plant cell will cause the chemical compositions to vary between micro-areas in phytolith. Cell lumen phytoliths might be expected to contain a lower carbohydrate content than cell wall phytoliths [26]. The EPMA of chemical compositions in phytolith is useful to ascertain and test these hypothesis and inferences.

The EPMA results of phytoliths provide the characteristic elemental compositions of phytoliths from different plant species, and therefore clarify the uncertainty arising from redundant production of phytolith morphotypes in distinct taxa. Thus, the EPMA of phytoliths might be helpful for identifying phytoliths, and could be used in archaeological plant tissue and paleobotany identification.

Elemental concentrations of phytoliths from plants differed between plants grown in polluted and unpolluted soils [51, 54]. The presence of Al in phytoliths might deduce they were produced by woody species [70]. Hart [67] postulated that the elemental content of phytoliths could be useful as a marker within plant, litter and soil systems. The phytolith formation in plants is affected not only by the plant species, but also the environmental conditions (temperature, moisture, pH, soil nutrition) of plant growth [4, 74]. It is possible to discern and discover

how the chemical compositions of phytoliths would respond to changes of environmental conditions [14]. Therefore, the EPMA of elemental compositions in phytoliths could be used to study environmental archeology and paleoenvironmental reconstruction.

## Conclusions

The direct analysis of C with other elements in phytoliths causes the abnormal high C content and total values in EPMA results. Non-carbon analysis and the Fix-C model were feasible in the EPMA of elemental compositions in phytolith. This Fix-C method was helpful for obtaining accurate compositions of other elements in phytoliths because C is involved in the ZAF correction. The NCA method was useful in distinguishing the elements of phytoliths with different morphologies.

The EPMA results showed that high $SiO_2$ content and small quantities of various other mineral elements compose phytoliths. Additionally, the $H_2O$ and C content could be calculated and estimated by using the Total values of EPMA. Preventing weathering and erosion of phytoliths and increasing the amounts of phytoliths in a polished section can improve the EPMA results. The EPMA of phytoliths can provide the elemental composition of micro-areas of an individual phytolith particle. The elemental compositions of phytolith varied with their morphotypes, the genera and ecotype of bamboos. EPMA of elemental compositions in phytolith is a potential tool to study the mechanism of phytolith formation, plant taxonomical identification, archaeological, paleoenvironmental and paleo-ecotype reconstruction.

## Supporting information

**S1 Table. EPMA results.** A. EPMA (using NCA and Fix-C method) results (elemental compositions and total values (wt.%)) of different phytolith morphotypes in leaves of bamboo species from three genera. **B.** Average elemental compositions and total values (wt.%) (using Fix-C and NCA method) of phytoliths in leaves of bamboo species from three genera. **C.** The elemental content of dominant phytolith morphologies from Dendrocalamus ronganensis leaves determined by EPMA with three methods.
(DOCX)

## Acknowledgments

We are grateful to Wanyi Liang and Siming Zhao for help in EPMA experiments.

## Author Contributions

**Conceptualization:** Rencheng Li.

**Data curation:** Shuhui Tan.

**Investigation:** Shuhui Tan, Xinyue Tao, Mengdan Wen, Haiyan Dong.

**Methodology:** Shuhui Tan, Yizhi Liu, Lintong Zhou.

**Project administration:** Rencheng Li.

**Supervision:** Rencheng Li.

**Writing – original draft:** Shuhui Tan.

**Writing – review & editing:** Shuhui Tan, Richard S. Vachula, Xinyue Tao, Mengdan Wen.

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
