## [Decision Letter · Decision Letter 0]

12 Apr 2022

PONE-D-22-02015Electron probe microanalysis of the elemental composition of phytoliths from woody bamboo speciesPLOS ONE

Dear Dr. Li,

Thank you for submitting your manuscript to PLOS ONE. After careful consideration, we feel that it has merit but does not fully meet PLOS ONE’s publication criteria as it currently stands. Therefore, we invite you to submit a revised version of the manuscript that addresses the points raised during the review process.

We look forward to receiving your revised manuscript.

Kind regards,

Alejandro Fernandez-Martinez, PhD

Academic Editor

PLOS ONE

Journal Requirements:

(This work is supported by grants from the National Natural Science Foundation of China (Grant Nos. 41867058).)

(This information should be included in your cover letter; we will change the online submission form on your behalf."

The following statement is declared in regard to our manuscript submission of Electron probe microanalysis of the elemental composition of phytoliths from woody bamboo species: 

All authors have read and approved this version of the article, and due care has been taken to ensure the integrity of the work. No part of this paper has been published or submitted elsewhere. No conflict of interest exists in the submission of this manuscript.)

Additional Editor Comments:

Dear Dr. Li

I have received the reviews by two experts in the field. They are of the opinion that your manuscript could be published at PlosONE after some revisions are performed. Whereas I share their opinion, I would like to encourage you to revise the English of the text.

I am looking forward to receiving a revised version of the manuscript.

Sincerely

Alejandro Fernandez-Martinez

Reviewers' comments:

Reviewer's Responses to Questions

**Comments to the Author**

1. Is the manuscript technically sound, and do the data support the conclusions?

Reviewer #1: Yes

Reviewer #2: Yes

2. Has the statistical analysis been performed appropriately and rigorously? 

Reviewer #1: Yes

Reviewer #2: Yes

3. Have the authors made all data underlying the findings in their manuscript fully available?

Reviewer #1: Yes

Reviewer #2: Yes

4. Is the manuscript presented in an intelligible fashion and written in standard English?

Reviewer #1: No

Reviewer #2: Yes

5. Review Comments to the Author

Reviewer #1: This paper report an interesting research on the electron probe microanalysis of

elemental compositions inside the bamboo phytoliths mounted in the slide. This research is really useful to the plant ecology and paleoenvironmental reconstruction.

Before consideration for publication, some questions should be resolved as the followings:

L35: what is tree genera?

L76: 10.12%, 3.37%

L77: PhytOC should have a complete description as first occurrence.

L218, 3.1

The text description is less frequent, and it needs to be strengthened.

L235, 237, Tables 2-3

The chemical composition content lacks units.

L391, 4.3

The author should add some figure and table to show the differences between phytolith morphotypes.

L412, 4.4

In present, there is no evident that the elemental compositions of phytoliths have taxonomical significance. Can the authors provide more evidences including the measurements of long saddles, etc. ? Otherwise, I suggest the authors delete the relative content.

The format of the reference needs to be corrected.

There are some linguistic errors in the text. It should be polished by an English native speaker.

Reviewer #2: I really appreciate Editor to invite me to review this manuscript written by Tan et al. After a careful reading, this manuscript is quite an original paper looking at the elemental composition of phytoliths from woody bamboo species using Electron probe microanalysis.

It relatively well-written, and well-organized, and systematized, presenting some original results, offering interesting findings referring the formation mechanism of phytoliths using a new technique.

Overall I support publication of this work, yet I have some comments to be considered (moderate revisions).

Figure 5 the decimal places of total values and SiO2 content are different for figure A,B,C. two decimal places are advised.

Line 27-28. “slide mounted” should be “mounted slide”.

Line 35 “tree genera” should be “three genera”

Line 52 “phytolith are normally….”, should be “phytolith morphology is normally”

Line 52. “cells” should be “cell”.

Line 55 “soil and sediments” keep the plural consistent

Line 59 delete the second “to”

Line 73. “lumen phytolith” should be “lumen phytoliths”.

Line 77 change “some research indicates” to “some researches indicate”

Line 89 “the elemental composition of micro-areas in individual phytolith particles”, “composition” should be in plural.

Line 92 “are” should be changed to “is”

Line 98 “are” should be changed to “is”

Line 108. “phytolith” should be “phytoliths”.

Line 112 (iii) What are the characteristic elemental compositions 113 in phytolith is difficult to understand.

Line 162. “10 min” should be “10 mins”.

Line 170. “10 min” should be “10 mins”.

Line 203. insert "phytolith particles included" after "20-30".

Line 207 “Dendrocalamus ronganensis” should be in abbreviation, please check through the paper

Line 219 “phytolith particle were”, change “were” to “was”

Line 231 “phytolith morphologies” should be changed to “phytolith morphotypes”

Line 235 Use phytolith morphotypes

Line 254 “all of Total carbon values” , delete “carbon”, Total should be changed to “total”

Line 282 Please check the use of “CO2”

Line 352 “The EPMA of elemental composition in micro-areas of phytolith in mounted slide are …” can be rephrased as “the EPMA results of phytolith in mounted slide are…”

Line 365 to 366, “from these two plant species”, not from two species, use “measured” is Ok

Line 373. “First” should be “Firstly”.

Line 451 change “morphologies” to “morphotypes”

Line 459 change “composition” to “compositions”

Line 464. change " distinguishe" to " to distinguish".

Supplemental data

S1 table the caption “the EPMA of phytolith….”should be change to “EPMA results of phytolith”

Table format looks not good.

S2 table the caption “the average elemental compositions”, “the” should be removed

S3 table the data should be present according to three EPMA methods.

6. PLOS authors have the option to publish the peer review history of their article (what does this mean?). If published, this will include your full peer review and any attached files.

Reviewer #1: No

Reviewer #2: No

---

## [Author Response · Author response to Decision Letter 0]

26 Apr 2022

Reponses to reviewers and editors

Reviewer #1: This paper report an interesting research on the electron probe microanalysis of elemental compositions inside the bamboo phytoliths mounted in the slide. This research is really useful to the plant ecology and paleoenvironmental reconstruction. Before consideration for publication, some questions should be resolved as the followings:

1. L35: what is tree genera?

Reply: This was a spelling mistake. “tree genera” should be “three genera”.

2. L76: 10.12%, 3.37%

Reply: We have made these changes.

3. L77: PhytOC should have a complete description as first occurrence.

Reply: Ok. The complete description of "PhytOC" is "phytolith-occluded carbon".

4. L218, 3.1. The text description is less frequent, and it needs to be strengthened.

Reply: We have removed the word “variation” from this section title.

5. L235, 237, Tables 2-3. The chemical composition content lacks units.

Reply: We have added this in table captions.

6. L391, 4.3. The author should add some figure and table to show the differences between phytolith morphotypes.

Reply: We have provide the Figure 3, table 2 and S1 Table in manuscript, which does exhibit the differences between the morphotypes.

7. L412, 4.4. In present, there is no evident that the elemental compositions of phytoliths have taxonomical significance. Can the authors provide more evidences including the measurements of long saddles, etc.? Otherwise, I suggest the authors delete the relative content.

Reply: Thanks for this comment. We have removed the relative content of using the relative abundances of elemental compositions of phytolith to distinguish bamboo species at the genus level, and now only state that the relative abundances of elemental compositions in phytoliths do not show taxonomical significance at the genus level. However, the elemental compositions of phytoliths from these bamboo leaves have ecotypic taxonomical significance in this study. Carnelli et al. investigated the chemical composition of phytoliths from 20 species occurring in subalpine and alpine grasslands, heaths, and woodlands on siliceous bedrock, and found that only woody species produced a high proportion of phytoliths containing aluminum. Thus, we believe the elemental compositions probably do distinguish some plant species, and may have taxonomical significance at the family and/or subfamily level. 

8. The format of the reference needs to be corrected.

Reply: We have revised them.

9. There are some linguistic errors in the text. It should be polished by an English native speaker.

Reply: The manuscript has been polished by an English native speaker.

Reviewer: 2

1. Figure 5 the decimal places of total values and SiO2 content are different for figure A,B,C. two decimal places are advised. 

Reply: We have changed the format of these numbers.

2. Line 27-28. “slide mounted” should be “mounted slide”.

Line 35 “tree genera” should be “three genera”

Line 52 “phytolith are normally….”, should be “phytolith morphology is normally”

Line 52. “cells” should be “cell”.

Line 55 “soil and sediments” keep the plural consistent

Line 59 delete the second “to”

Line 73. “lumen phytolith” should be “lumen phytoliths”.

Line 77 change “some research indicates” to “some researches indicate”.

Line 89 “the elemental composition of micro-areas in individual phytolith particles”, “composition” should be in plural.

Line 92 “are” should be changed to “is”

Line 98 “are” should be changed to “is”

Line 108. “phytolith” should be “phytoliths”.

Reply: We have made these revisions.

3. Line 112 (iii) What are the characteristic elemental compositions 113 in phytolith is difficult to understand.

Reply: We have changed this sentence as “What are the EPMA results of a phytolith particle?”.

4. Line 162. “10 min” should be “10 mins”.

Line 170. “10 min” should be “10 mins”.

Line 203. insert "phytolith particles included" after "20-30".

Line 207 “Dendrocalamus ronganensis” should be in abbreviation, please check through the paper.

Line 219 “phytolith particle were”, change “were” to “was”

Line 231 “phytolith morphologies” should be changed to “phytolith morphotypes”

Line 235 Use phytolith morphotypes

Line 254 “all of Total carbon values”, delete “carbon”, Total should be changed to “total”

Line 282 Please check the use of “CO2”

Line 352 “The EPMA of elemental composition in micro-areas of phytolith in mounted slide are …” can be rephrased as “the EPMA results of phytolith in mounted slide are…”

Line 365 to 366, “from these two plant species”, not from two species, use “measured” is Ok

Line 373. “First” should be “Firstly”.

Line 451 change “morphologies” to “morphotypes”

Line 459 change “composition” to “compositions”

Line 464. change " distinguishe" to " to distinguish".

Reply: We have made these corrections.

5. Supplemental data

S1 table the caption “the EPMA of phytolith….”should be change to “EPMA results of phytolith”

Table format looks not good.

S2 table the caption “the average elemental compositions”, “the” should be removed.

S3 table the data should be present according to three EPMA methods.

Reply: Thanks for your comments. We have made these revisions.

---

## [Decision Letter · Decision Letter 1]

20 Jun 2022

Electron probe microanalysis of the elemental composition of phytoliths from woody bamboo species

PONE-D-22-02015R1

Dear Dr. Li,

We’re pleased to inform you that your manuscript has been judged scientifically suitable for publication and will be formally accepted for publication once it meets all outstanding technical requirements.

Kind regards,

Andrew W. Rate, Ph.D.

Academic Editor

PLOS ONE

Additional Editor Comments (optional):

The authors have done a good job of revising this manuscript in response to the comments of two expert reviewers.

Reviewers' comments:

Reviewer's Responses to Questions

**Comments to the Author**

1. If the authors have adequately addressed your comments raised in a previous round of review and you feel that this manuscript is now acceptable for publication, you may indicate that here to bypass the “Comments to the Author” section, enter your conflict of interest statement in the “Confidential to Editor” section, and submit your "Accept" recommendation.

Reviewer #1: All comments have been addressed

Reviewer #2: All comments have been addressed

2. Is the manuscript technically sound, and do the data support the conclusions?

Reviewer #1: Yes

Reviewer #2: Yes

3. Has the statistical analysis been performed appropriately and rigorously? 

Reviewer #1: Yes

Reviewer #2: Yes

4. Have the authors made all data underlying the findings in their manuscript fully available?

Reviewer #1: Yes

Reviewer #2: Yes

5. Is the manuscript presented in an intelligible fashion and written in standard English?

Reviewer #1: Yes

Reviewer #2: Yes

6. Review Comments to the Author

Reviewer #1: The authors modify the ms according to the reviewers suggestions. The ms is better than before. In present, it is suitable for publication on PLOS ONE. Pls correct all "Fig" as "Fig.".

Reviewer #2: This response gives me full satisfaction, as authors already did in revised manuscript.

Congrats to all authors on this nice work.

7. PLOS authors have the option to publish the peer review history of their article (what does this mean?). If published, this will include your full peer review and any attached files.

Reviewer #1: No

Reviewer #2: No

---

## [Editor Report · Acceptance letter]

24 Jun 2022

PONE-D-22-02015R1 

Electron probe microanalysis of the elemental composition of phytoliths from woody bamboo species 

Dear Dr. Li:

I'm pleased to inform you that your manuscript has been deemed suitable for publication in PLOS ONE. Congratulations! Your manuscript is now with our production department. 

Kind regards, 

on behalf of

Dr. Andrew W. Rate 

Academic Editor

PLOS ONE